# Benchmarking Few-shot Transferability of Pre-trained Models with Improved Evaluation Protocols

## Abstract

Few-shot transfer has been made possible by stronger pre-trained models and improved transfer algorithms. However, there lack of a unified, rigorous evaluation protocol that is challenging yet meets real-world usage. To this end, we carefully review previous evaluation principles and establish new standards with recipes from different aspects following our empirical findings, including the report of confidence intervals, the standard for hyperparameter tuning, and variation of ways and shots, etc. With these standards, we create FewTrans, a few-shot transfer benchmark containing 10 challenging datasets from diverse domains with three sub-benchmarks: one that compares pre-trained models, one that compares transfer algorithms for vision-only models, and one that compares transfer algorithms for multimodal models. To facilitate future research, we reimplement and compare some of the recent pre-trained models and transfer algorithms. We observe that, while stronger pre-trained models bring significant performance improvement, the performance of most transfer methods is quite close, and simply finetuning the whole backbone performs well enough, especially for multi-modal models. We hope that the release of FewTrans benchmark will streamline reproducible and rigorous advances in few-shot transfer learning research.

## 1 Introduction

Recent progress on computer vision (Kolesnikov et al., 2020; Radford et al., 2021; Islam et al., 2021; Dehghani et al., 2023) suggests that good performance on a variety of vision tasks can be achieved at low cost by transferring a pretrained, large-scale model with only a few labeled samples, facilitating downstream scenarios where labeled data can be expensive or difficult to obtain. This *few-shot transferability* of pre-trained models can be further improved by adopting recently proposed transfer algorithms that are claimed to be better than vanilla finetuning in terms of accuracy or efficiency, such as partial finetuning (Zaken et al., 2022), low-rank adaptation (Hu et al., 2022), adapter tuning (Houlsby et al., 2019; Chen et al., 2022; Li et al., 2022b), meta-learning (Shysheya et al., 2023), prompt tuning (Jia et al., 2022; Zhou et al., 2022c; Khattak et al., 2023) and so on.

However, the evaluation criteria of few-shot transfer have not been unified and diverge across separate threads of research, which hinders newly proposed pretrained models or transfer algorithms from being accurately evaluated and compared with previous ones. To build a unified, reasonable evaluation protocol, we first review previous evaluation setups for few-shot transfer with careful experiments. We find several inappropriate aspects caused by the specific few-shot nature of few-shot transfer problems overlooked by previous evaluation criteria.

In particular, we find two major deficiencies in previous evaluation setups. First, we observe that a single few-shot task has large performance variation caused by random sampling of training data, thus previous reports of few-shot performance with few tasks are unreliable: just by changing seeds that generate tasks, one can obtain high performance with arbitrary methods. This problem can be handled easily by sampling more tasks. We then note that current hyperparameter selection criterion that sets an additional validation dataset from the target domain is not realistic for real-world few-shot problems. We argue that model selection should be either dependent on a dataset irrelevant to the target dataset, or should be only dependent on the downstream task at hand. Our further

analysis shows that the optimal hyperparameters of few-shot transfer change from task to task and from dataset to dataset, thus designing a reasonable model selection criterion that reflects the real performance of models/methods while being fair is difficult. We thus propose to use hyperparameter ensemble (Wenzel et al., 2020) which avoids looking for a single hyperparameter but instead classifies test samples using several adapted classifiers obtained from a range of hyperparameters.

Integrating all our solutions to major and minor deficiencies of previous evaluation protocols, we construct FEWTRANS, a few-shot transfer benchmark containing 10 diverse downstream datasets, with the ability of sampling class-imbalanced tasks with varying numbers of classes and shots. FEWTRANS has three sub-benchmarks for comparing pretrained models, transfer algorithms for vision-only models, and transfer algorithms for multimodal models respectively. To facilitate future research, we have reimplemented and compared a bunch of pretrained models and transfer algorithms. We have several interesting observations. We observe that while a larger pretraining dataset contributes significantly to the downstream few-shot performance, different transfer algorithms have quite close performance. A simple all-parameter finetuning performs surprisingly well and seems not to meet overfitting problems especially for multimodal models, calling into question whether we are making progress on the problem.

## 2 RELATED WORK

Few-shot transferability of pre-training models improves with larger training datasets, architectures, and better pre-training algorithms. Kornblith et al. (2019) verify that models transferred from supervised ImageNet models generally perform much better than those trained from scratch on downstream tasks, especially under few-shot settings. Self-supervised ImageNet models were then shown to be better source models on few-shot transfer tasks than supervised models (Islam et al., 2021; Luo et al., 2023). Recent studies (Kolesnikov et al., 2020; Zhai et al., 2022; Dehghani et al., 2023) further show that scaling up pre-training datasets to hundreds of millions and parameters to billions leads to stably increasing few-shot transfer performance. On the other hand, the CLIP model (Radford et al., 2021) leverages multimodal data for pre-training and achieves very impressive zero/few-shot performance on a suit of visual classification datasets using hand-crafted text prompts.

Unlike many-shot transfer learning literature where standard benchmarks like VTAB (Zhai et al., 2019) exist, most papers that evaluate few-shot transferability of pretrained models do not use benchmarks but instead self-select datasets for evaluation (Kornblith et al., 2019; Kolesnikov et al., 2020; Radford et al., 2021). One exception is the few-shot transfer benchmark of transfer algorithms for multimodal models (Zhou et al., 2022c) that has 11 downstream datasets. Some evaluation principles of our benchmark were largely inspired by Meta-Dataset (Triantafillou et al., 2020), a benchmark for classical few-shot classification problems. There are several reasons for why we do not build our benchmark on the top of Meta-Dataset, including no class names in some datasets, unnatural image preprocessing in some datasets, having too many shots in a task, etc.

## 3 THE PROBLEM OF FEW-SHOT TRANSFER LEARNING

In transfer learning, we have a pretrained model $f_\theta : \mathbb{R}^d \to \mathbb{R}^m$ mapping inputs $x \in \mathbb{R}^d$ to features $z \in \mathbb{R}^m$. The goal of transfer learning is to transfer the pretrained model $f_\theta$ to a specified downstream task. Any downstream task $\tau$ can be described as a combination of a training set $D^{tr} = \{(x_i, y_i)\}_{i=1}^N$ and a test set $D^{te} = \{(x_j^*, y_j^*)\}_{i=1}^M$, where $y_i, y_j^* \in \{1, ..., n_{cls}\}$ are class labels. The task $\tau$ is called a $K$-shot task if there are exactly $K$ samples per class in $D^{tr}$. During transfer, the pretrained model $f$ will be adapted to task $\tau$ using the training set $D^{tr}$ through a transfer algorithm such as finetuning, producing a new classifier mapping images to labels of the new task. To evaluate the effectiveness of transfer, the produced classifier will be evaluated on the test set $D^{te}$.

In a typical transfer learning evaluation setup (Zhai et al., 2019), the downstream task involves an entire downstream dataset, so the number of samples per class can be quite large, deviating from some practical transfer scenarios where downstream data is difficult to obtain. Under few-shot transfer scenario, the number of samples per class can be quite small, usually less than 20 or 10.

## 4 INAPPROPRIATE EVALUATION OF PREVIOUS METHODS

In this section, we point out several flaws of previous few-shot transfer evaluation protocols. For all experiments done in this section, we use fine-tuning as the transfer algorithm. Following Luo et al. (2023), we separately set the learning rates for the backbone of the pretrained model and the linear head for improved performance. By default, we use Adam (Kingma & Ba, 2015) as the optimizer.

### 4.1 LARGE PERFORMANCE FLUCTUATION CAUSED BY SAMPLING

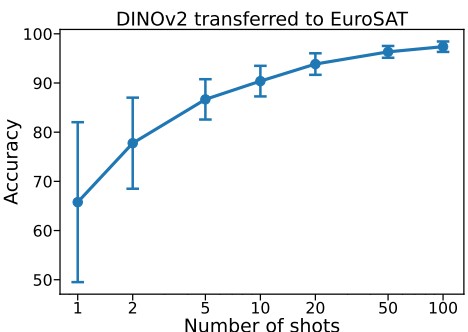

Figure 1: Average accuracy and 95% confidence intervals of single-task few-shot transfer evaluation of a pretrained DINOv2-small model on EuroSAT (Helber et al., 2019).

Different from the typical transfer learning setup where an entire dataset is used as the downstream task, in few-shot transfer, a randomly sampled small part of the dataset is used as the downstream task. Previous works that evaluate pretrained models on few-shot transfer tasks (Kolesnikov et al., 2020; Radford et al., 2021; Zhou et al., 2022c) sample a single or a few (usually 3) tasks and only report the average performance on the sampled tasks without error bars. This can be problematic because the performance can be largely influenced by the choice of sampled data especially under few-shot settings. To illustrate this, we give the single-task transfer performance of DINOv2-small (Oquab et al., 2023) on the EuroSAT dataset (Helber et al., 2019) along with the 95% confidence intervals in Figure 1. As seen, when the number of shots is small, the spread of the error bar can be very large. For 1-shot task, the performance can vary from less than 50% to more than 80% within the confidence interval. This is caused by the randomness of the training set, where a change to a single sample can lead to large fluctuations in performance (Agarwal et al., 2021). Thus the comparison in previous works using only a few tasks is inappropriate because the change of seed can determine the rank of pretrained models/transfer methods completely. To make the comparison meaningful, we should at least sample more tasks to make the confidence interval small enough.

### 4.2 UNREALISTIC MODEL SELECTION

In the typical transfer learning setting, the downstream dataset is so large that we can partition it into a training set for adptation, and a validation set for selecting hyperparameters like learning rates and number of epochs for adaptation. When it comes to the evaluation of few-shot transferability of pretrained models, previous works either tune hyperparameters on a large validation set (possibly from different classes) from the same dataset (Radford et al., 2021; Luo et al., 2023) or set hyperparameters to some default "magic" values dependent on downstream datasets (Kolesnikov et al., 2020; Li et al., 2022b; Xu et al., 2022; Zhou et al., 2022c). While it seems valid to tune hyperparameters on a separated validation set as what is done in the traditional many-shot transfer learning literature, we point out that doing so is inappropriate in the few-shot setting because it deviates from real-world scenarios where additional labeled data from the same dataset for validation is hard to obtain.

Thus to make the evaluation protocol realistic while being fair for comparison, we have two choices: (1) determine hyperparameters of transfer algorithms in advance on a *held-out dataset* that is both different from the pretraining dataset and target downstream dataset; (2) determine hyperparameters based on the few training samples of the target downstream dataset on the fly. We will next evaluate the validity of these two choices.

**Optimal hyperparameters change from task to task.** If we determine hyperparameters on a separate dataset, then the hyperparameters will be the same for all tasks. Is this appropriate? In Table 1, we show the optimal hyperparamters of ten tasks sampled from the same dataset. We can observe that, even when sampled from the same dataset with the same set of classes, tasks with different training samples can have different optimal hyperparamters. The optimal number of epochs varies from $15$ to $40$; the optimal learning rate for pretrained backbone varies from $2e - 06$ to $5e - 05$; the optimal learning rate for linear classifier varies from $0.01$ to $0.2$.

Table 1: Optimal hyperparameters vary from task to task. The pretrained model is DINOv2-small, and all tasks are 1-shot sampled from EuroSAT.

| Task ID | 0 | 1 | 2 | 3 | 4 | 5 | 6 | 7 | 8 | 9 |
|---|---|---|---|---|---|---|---|---|---|---|
| Epoch | 15 | 15 | 15 | 40 | 15 | 40 | 30 | 20 | 20 | 30 |
| Backbone lr | 5e-05 | 5e-06 | 5e-06 | 1e-05 | 2e-06 | 5e-06 | 1e-05 | 2e-05 | 2e-05 | 1e-05 |
| Head lr | 0.05 | 0.01 | 0.2 | 0.02 | 0.01 | 0.05 | 0.01 | 0.05 | 0.05 | 0.2 |

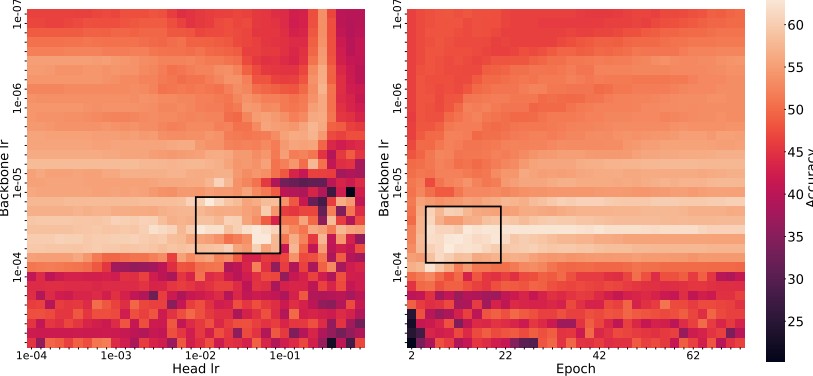

Figure 2: The heatmaps showing how the few-shot transfer performance of a single 1-shot task sampled from EuroSAT changes with hyperparameters. We fix the number of epochs to 50 in the left plot, and fix the head lr to 0.01 in the second plot. The black rectangles highlight the optimal hyperparameter areas.

**Few-shot transfer performance is sensitive to the choice of hyperparameters.** Only showing that the optimal hyperparameters change from task to task is not enough to conclude that the few-shot transfer performance will change from task to task if we use the same hyperparameters for all tasks. We still need to show that few-shot transfer performance is sensitive to hyperparameters. We show how sensitive few-shot transfer performance is to the choice of hyperparameters in Figure 2. We plot the heatmaps of few-shot transfer performance of a single 1-shot task when varying two of the hyperparameters. As we can see, the variation of accuracy can be very large in the considered ranges, from around 20% to more than 60%. In particular, the performance can drop very quickly when we move out of the optimal area (highlighted in the black rectangle). For example, in the left plot, if we go down or right from the black rectangle, that is, increasing the learning rate of the backbone or the linear head, we will go into a chaotic area, where the accuracy oscillates up and down irregularly and often drops to half or even less. This phenomenon seems not that evident for the number of epochs in the right plot where the accuracy seems to be smoother, but we can still see a 10% performance fluctuation around the optimal area.

**Optimal hyperparameters change from dataset to dataset.** Even if we can tolerate the performance variation per task, we show that the "average optimal hyperparameters"—the hyperparameters that give the highest average performance over several tasks sampled from a dataset—can still vary from dataset to dataset in Table 2. For example, the optimal number of epochs when transferred to Plant Disease (Mohanty et al., 2016) is 50, while the optimal number of epochs when transferred to UCF101 (Soomro et al., 2012) is 10. Among the six downstream datasets, the backbone learning rate ranges from $1e-06$ to $2e-05$, and the head learning rate ranges from $5e-04$ to $1e-02$.

Combining the analysis above, we can conclude that single hyperparameters will lead to unstable few-shot transfer performance from task to task and from dataset to dataset. So this hyperparameter selection criteria will cause large uncertainty of few-shot transfer performance and thus cannot reflect the true performance of different methods. Thus a proper hyperparameter selection criteria should rely only on the training set of the downstream dataset at hand, which we will explore next.

**Cross-validation fails to provide reliable estimation of hyperparameters.** A representative way of estimating the hyperparameters using the training set of downstream tasks is *cross-validation* which has a long history of use in machine learning (Kohavi et al., 1995; Arlot & Celisse, 2009). The main idea behind $l$-fold cross-validation is to split data $l$ times, each time into a training part

Table 2: Average optimal hyperparameters of few-shot transfer vary from dataset to dataset. The pretrained model is DINOv2-small, and all tasks are 1-shot.

|  | CIFAR-100 | UCF | Plant Disease | Aircraft | DTD | EuroSAT |
|---|---|---|---|---|---|---|
| Epoch | 30 | 10 | 50 | 30 | 20 | 30 |
| Backbone lr | 1e-05 | 1e-05 | 2e-05 | 1e-06 | 1e-05 | 1e-05 |
| Head lr | 0.0005 | 0.01 | 0.005 | 0.005 | 0.001 | 0.01 |

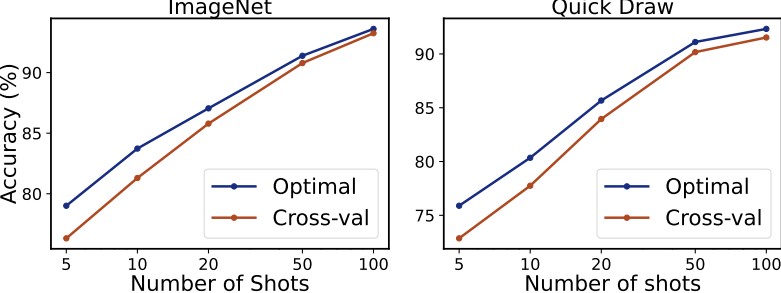

Figure 3: Cross-validation cannot find good hyperparameters when the number of shots is small, regardless of the domain shift between pretraining and downstream dataset. We use a subset class of ImageNet as the training set, and use the remaining part as the downstream dataset for the left plot.

and a validation part. For each time of split, the training part is used to adapt the model and the validation part is used to evaluate the adaptation. The hyperparameters are chosen such that the average error over all splits is small. While cross-validation can work well when there is abundant data, we find that it will meet difficulties when data for adaptation is scarce, because (1) the number of samples per class is too small to split. For example, when the number of samples is below 5, it is only possible to use the leave-one-out strategy, that is, the validation part only has 1 sample for each spilt, leading to unreliable performance estimation. For extreme 1-shot case, we cannot apply cross-validation because there is no data to split; (2) $l$-fold cross-validation changes the number of shots from $K$ to $K(l-1)/l$. As we have shown previously, optimal hyperparameters for few-shot transfer can change when the task has changed, thus the hyperparameters found by cross-validation can be biased. We verify our considerations in Figure 3, where we show that there is a gap between the accuracy obtained by 5-fold cross-validation and the accuracy obtained by using the "average optimal hyperparameters" of the dataset for both in-domain and out-of-domain transfer, especially when the number of shots is small.

In conclusion, figuring out the optimal hyperparameters for few-shot transfer is very important and is, if not impossible, very difficult under real-world settings. Because of this difficulty, a good few-shot transfer method should not only have high performance at its optimal hyperparameters, but should also have resistance to the change of hyperparameters, that is, the test loss landscape around the optimal hyperparameters should be flat such as we can tolerate an inevitable deviation of hyperparameter estimations. Thus a good evaluation protocol should evaluate both the performance that a pretrained model can reach, as well as its sensitivity to the choice of hyperparameters.

### 4.3 OTHER CONSIDERATIONS

Apart from the aforementioned two major defects of previous evaluation of few-shot transferability of pretrained models, we also notice several other points that can be improved, with some of them inspired by the few-shot learning literature (Triantafillou et al., 2020).

**No variation of the number of classes.** Following transfer learning literature, papers that evaluates the few-shot transferability of pretrained models often use all classes of the target downstream dataset (Kolesnikov et al., 2020; Radford et al., 2021) to form a task, thus the capability of pretrained models transferring to less number of classes which forms more specific fine-grained category structures is not considered.

**No class imbalance.** For simplicity, almost all previous few-shot transfer evaluations use class-balanced settings, where the number of shots in each class of the training set is exactly the same

Table 3: Average 1-shot transfer performance of pretrained DINOv2-small over 50 tasks: hyperparameter ensemble vs. individual hyperparameter configurations. See appendix for details.

| configuration | (1,1) | (1,2) | (1,3) | (2,1) | (2,2) | (2,3) | (3,1) | (3,2) | (3,3) | Avg | lr ensemble | lr+epoch ensemble |
|---|---|---|---|---|---|---|---|---|---|---|---|---|
| EuroSAT | 67.52 | 67.23 | 68.00 | 70.75 | 71.01 | 61.87 | 45.44 | 45.55 | 42.48 | 59.98 | 70.21 | 70.72 |
| Aircraft | 61.92 | 61.55 | 61.44 | 61.49 | 61.79 | 61.23 | 61.44 | 61.31 | 60.45 | 61.40 | 63.39 | 63.28 |

for all classes. However, we cannot guarantee that this will still hold in real-world few-shot transfer scenarios and thus models and algorithms should be evaluated on class-imbalanced scenarios.

**Datasets lack diversity, are too easy, and may have errors.** Take the widely-used few-shot transfer benchmark for multimodal pretrained models (Radford et al., 2021; Zhou et al., 2022c) that contains 11 datasets as an example. Images from most datasets in this benchmark are taken from modern cities, thus being similar to parts of ImageNet and the tasks are not difficult to solve even when there are only a few samples per class. This can be seen from recent papers (Khattak et al., 2023) where the average few-shot accuracy of 5 datasets reaches more than 90%, in the condition that some of the datasets have more than 100 classes, which should have been difficult to classify correctly with few samples per class. In addition, the benchmark has StanfordCars (Krause et al., 2013) as one of its datasets, which has proven to have tons of mislabeled images and outliers (Cleanlab, 2023).

## 5 Introducing the FewTrans Benchmark

In this section, we introduce several evaluation standards to solve the aforementioned issues of few-shot transfer, which constitutes the key components of our proposed FewTrans benchmark.

### 5.1 Hyperparameter Ensemble for Robust Few-shot Evaluation

To overcome the difficulty of estimating hyperparameters with a few samples, we propose to not search for single hyperparameters, but instead use hyperparameter ensemble (Momma & Bennett, 2002; Wenzel et al., 2020) that utilizes several hyperparameters for prediction. Specifically, let $\mathcal{H} = \{h^i\}_{i=1}^m$ be a set of $m$ hyperparameter configurations, where $h^i = \{h_1^i, h_2^i, ..., h_n^i\}$ is a single configuration that includes values of $n$ hyperparameters. Suppose that the classifier produced by adapting a pre-trained model to a downstream task by hyperparameter configuration $h^i$ is $g_{h^i}$, which maps images to classification scores. Then for any given test image $x$ of the downstream task, the classification score of $x$ wrt hyperparameter ensemble $\mathcal{H}$ is defined as the sum of all scores obtained by each hyperparameter configuration, i.e., $\sum_{h \in \mathcal{H}} g_h(x)$.

One advantage hyperparameter ensemble offers is its robustness to individual bad hyperparameter configurations. We can observe this in the first row of Table 3: the accuracies of some hyperparameter configurations are very low (less than 50%), but the hyperparameter ensemble still reaches 70.21% accuracy, very close to the optimal performance 71.01% achieved by the optimal individual configuration. Thus as long as the good hyperparameters are included in $\mathcal{H}$, the few-shot transfer performance with hyperparameter ensemble will be very close to the performance obtained by the good hyperparameters. As we have seen before, the variations of hyperparameters of a given pretrained model are not beyond several orders, thus as long as we set a large enough range of hyperparameters, every evaluated task can provably approach its optimal performance and the evaluation is thus stabilized to some extent. In addition, it does not introduce additional computation overhead compared to cross-validation.

Another advantage of using hyperparameter ensemble is that it can measure the sensitivity of the few-shot transfer performance to the choice of hyperparameters. As seen from the second row of Figure 3, when the loss landscape around the optimal hyperparameters is flat, the performance given by the ensemble will be higher, while not causing too strong fluctuations.

According to what we have just discussed, the two criteria that we require for a good hyperparameter searcher are both satisfied by hyperparameter ensemble. We thus use it in FewTrans. For practical usage, we still need to determine how to set the range of hyperparameters for each pretrained models/transfer algorithm. Since we know that the hyperparameters won't usually change too much from dataset to dataset, we determine the range on a held-out dataset by finding the best average

Table 4: Sub-benchmark of FEWTRANS that compares the few-shot transferability of different pre-trained models. We use all-parameter finetune as the transfer algorithm for all models. We temporarily do not evaluate pretrained models that use larger architectures.

| Models | Dataset | ImageNet-S | DTD | CIFAR-100 | Flowers | UCF | EuroSAT | Quick Draw | Fungi | Plant Disease | Aircraft | Average |
|---|---|---|---|---|---|---|---|---|---|---|---|---|
| ResNet-50 | IN-1M | 63.6±1.5 | 69.3±1.1 | 74.3±1.1 | 84.1±1.1 | 76.7±1.2 | 84.1±1.0 | 64.8±1.3 | 47.6±1.4 | 72.5±1.4 | 51.7±1.4 | 68.9±1.3 |
| MAE-base | IN-1M | 72.3±1.5 | 74.0±1.1 | 86.3±0.8 | 92.1±0.6 | 88.2±0.8 | 88.1±0.9 | 72.6±1.2 | 62.1±1.3 | 85.9±0.8 | 52.3±1.3 | 77.4±1.1 |
| Swin-B | IN-1M | 71.5±1.5 | 74.7±1.1 | 87.0±0.8 | 90.9±0.8 | 87.8±0.8 | 87.6±0.9 | 74.2±1.1 | 60.7±1.3 | 86.1±0.8 | 55.4±1.4 | 77.6±1.1 |
| EsViT-SwinB | IN-1M | 69.7±1.5 | 75.3±1.0 | 84.7±0.9 | 93.6±0.6 | 88.0±0.8 | 87.8±0.9 | 70.8±1.2 | 62.9±1.3 | 86.2±0.8 | 57.7±1.5 | 77.7±1.1 |
| ConvNext-B | IN-1M | 74.6±1.5 | 73.9±1.1 | 88.3±0.7 | 90.8±0.8 | 89.1±0.8 | 86.4±1.0 | 75.1±1.1 | 60.8±1.3 | 84.2±0.9 | 55.7±1.4 | 77.9±1.1 |
| IBOT-ViT | IN-1M | 69.7±1.5 | 74.9±1.1 | 89.6±0.7 | 93.6±0.6 | 89.1±0.8 | **89.1**±0.8 | 69.9±1.2 | 60.8±1.3 | 86.3±0.8 | 56.7±1.5 | 78.0±1.1 |
| BiT-R101 | IN-14M | 68.2±1.5 | 77.3±1.0 | 85.3±0.8 | 99.6±0.1 | 89.4±0.7 | 87.0±0.9 | 71.9±1.2 | 67.9±1.2 | 91.1±0.6 | 57.5±1.4 | 79.5±1.0 |
| CLIP-base | WIT-400M | **80.4**±1.3 | **83.5**±0.8 | **94.9**±0.3 | 96.9±0.3 | **95.4**±0.3 | 88.5±0.7 | 76.5±0.9 | 54.8±1.4 | 77.2±1.0 | **78.8**±1.0 | 82.7±0.9 |
| DINOv2-small | LVD-142M | 75.1±1.4 | 81.3±0.9 | 89.8±0.7 | 99.6±0.1 | 90.3±0.7 | 87.0±0.9 | 78.1±1.0 | 69.9±1.2 | 89.8±0.7 | 67.3±1.4 | 82.8±1.0 |
| DINOv2-base | LVD-142M | 79.8±1.3 | 82.6±0.9 | 93.0±0.5 | **99.9**±0.0 | 93.9±0.5 | 87.7±0.9 | **79.6**±1.0 | **74.9**±1.1 | **91.8**±0.6 | 70.3±1.3 | **85.3**±0.9 |

optimal hyperparameters on this dataset, setting it as the center of the hyperparameter range, and expanding it to a full range. For pretrained models not trained on ImageNet, we choose the validation set of ImageNet as the held-out dataset, while for ImageNet models, we choose CUB (Welinder et al., 2010) as the held-out dataset.

## 5.2 OTHER COMPONENTS OF FEWTRANS

**Datasets.** We choose datasets such that the sampled tasks are not too easy, cover different domains, and do not have many errors. In addition, in order to evaluate multimodal models, we require that each class of chosen datasets should have a text name. We finally choose ten datasets that satisfy these criteria: ImageNet-Sketch (Wang et al., 2019), DTD (Cimpoi et al., 2014), CIFAR-100 (Krizhevsky et al., 2009), VGG Flowers (Nilsback & Zisserman, 2008), UCF-101 (Soomro et al., 2012), EuroSAT (Helber et al., 2019), Quick Draw (Jonas et al., 2016), Fungi (Schroeder & Cui, 2018), Plant Disease (Mohanty et al., 2016) and Aircraft (Maji et al., 2013).

**Base-novel split.** Following literature of transfer algorithms for multimodal models (Zhou et al., 2022c; Khattak et al., 2023), we split the classes of each dataset into a base set of classes and a novel set of classes. For base evaluation, the pretrained multimodal model will be adapted to the training set sampled from base set and evaluated on the test set sampled from base set. For base-to-novel evaluation, the pretrained multimodal model will still be adapted to the training set sampled from the base set, but evaluated on the test set sampled from the novel set of classes. This is possible since multimodal models like CLIP (Radford et al., 2021) do not need a tunable classification head, but classify images dependent on text names of classes only. For unimodal models, we only conduct base evaluation. The base-novel split is approximately $4 : 1$ for each dataset.

**Sampling criteria.** We follow the task sampling criteria adopted in Meta-Dataset (Triantafillou et al., 2020) with some small differences. Specifically, to sample a task, we first sample a random number of classes from the target task. The number of classes is sampled uniformly from $[2, 15]$ for all datasets except for ImageNet-Sketch, whose classes per task are hierarchically sampled from one node in WordNet to improve the quality of sampled tasks. Then images in the task are sampled with an imbalance of shots for each class. In Meta-Dataset, the average number of shots can be large (20 or more), deviating from the true few-shot settings. We thus restrict the maximum number of training samples in each class to 10, constructing "true" few-shot tasks. To have a well estimation of performance, we sample 600 tasks per dataset and report the 95% confidence intervals.

## 5.3 EXPERIMENTS ON FEWTRANS

We use the aforementioned evaluation protocols to evaluate the few-shot transferability of pretrained models and compare different transfer algorithms. This results in three sub-benchmarks that (1) compares different pretrained models, (2) compares different transfer algorithms for pure vision

Table 5: Sub-benchmark of FEWTRANS that compares different transfer algorithms for pure vision pretrained models. The visual encoder of CLIP-base is chosen as the pretrained model.

| | ImageNet-S | DTD | CIFAR-100 | Flowers | UCF | EuroSAT | Quick Draw | Fungi | Plant Disease | Aircraft | Average |
|---|---|---|---|---|---|---|---|---|---|---|---|
| Linear | 72.1±1.5 | 76.7±1.1 | 83.7±0.8 | 95.5±0.4 | 91.6±0.7 | 81.5±1.0 | 70.8±1.1 | 56.8±1.4 | 75.3±1.1 | 68.0±1.3 | 77.2±1.1 |
| Finetune | 73.1±1.5 | 79.9±1.0 | 88.0±0.8 | 95.9±0.5 | 93.0±0.6 | 87.6±0.9 | **78.9**±1.0 | 58.9±1.4 | 83.7±0.9 | 70.7±1.3 | 81.0±1.0 |
| LoRA | 73.8±1.5 | 80.7±1.0 | 88.7±0.7 | 96.1±0.4 | 93.3±0.6 | **87.7**±0.9 | 78.0±1.1 | 59.4±1.4 | 83.3±0.9 | 71.0±1.3 | 81.2±1.0 |
| BitFit | 73.6±1.5 | 79.7±1.0 | 89.3±0.7 | 96.8±0.4 | 93.3±0.6 | 86.5±0.9 | 77.3±1.1 | 61.3±1.3 | 83.5±0.9 | 71.0±1.2 | 81.2±1.0 |
| SSF | 74.2±1.5 | 80.3±1.0 | 89.0±0.7 | 96.7±0.4 | 93.2±0.6 | 87.4±0.9 | 77.3±1.1 | 60.8±1.3 | 84.4±0.9 | 70.7±1.3 | 81.4±1.0 |
| Adapter | 74.1±1.5 | 80.5±1.0 | 89.8±0.7 | 96.9±0.4 | 93.6±0.5 | 86.5±0.9 | 77.3±1.0 | 61.2±1.3 | 83.2±0.9 | 70.9±1.2 | 81.4±1.0 |
| Adaptformer | 74.1±1.5 | 80.8±1.0 | 90.0±0.7 | **97.0**±0.3 | **93.8**±0.5 | 87.0±0.9 | 77.7±1.0 | 61.8±1.3 | 83.6±0.9 | 71.0±1.2 | 81.7±1.0 |
| VPT | 73.2±1.5 | **82.1**±0.9 | **90.2**±0.7 | **97.0**±0.4 | 93.6±0.5 | 87.3±0.9 | 78.2±1.0 | **61.9**±1.3 | 85.7±0.9 | 71.6±1.2 | 82.1±1.0 |
| TSA | **74.3**±1.5 | 80.0±1.0 | 89.5±0.7 | 96.9±0.4 | 93.5±0.6 | 87.5±0.9 | 78.3±1.0 | 64.5±1.3 | **86.2**±0.8 | **72.2**±1.2 | **82.3**±1.0 |

Table 6: Sub-benchmark of FEWTRANS that compares different transfer algorithms for base evaluation of multi-modal pretrained models. CLIP-base is chosen as the pretrained model.

| | ImageNet-S | DTD | CIFAR-100 | Flowers | UCF | EuroSAT | Quick Draw | Fungi | Plant Disease | Aircraft | Average |
|---|---|---|---|---|---|---|---|---|---|---|---|
| Zero-shot | 72.6±1.5 | 73.0±1.0 | 92.9±0.4 | 86.3±0.9 | 90.5±0.6 | 64.4±1.2 | 57.4±1.2 | 38.7±1.5 | 46.0±1.4 | 69.2±1.2 | 69.1±1.2 |
| CoOp | 79.3±1.3 | 83.8±0.8 | 93.8±0.4 | 97.8±0.2 | 95.1±0.4 | 84.3±0.8 | 73.8±0.9 | 51.9±1.5 | 70.9±1.2 | 70.0±1.4 | 80.1±1.0 |
| ProGrad | 79.4±1.3 | 82.3±0.8 | 93.9±0.4 | 96.2±0.3 | 94.7±0.4 | 84.1±0.8 | 72.5±0.9 | 53.8±1.4 | 71.6±1.1 | 73.2±1.2 | 80.2±0.9 |
| VPT | 78.8±1.3 | 81.3±0.8 | 94.5±0.3 | 95.5±0.4 | 94.5±0.4 | 88.3±0.7 | 75.1±0.9 | 47.4±1.5 | 72.9±1.1 | 76.5±1.1 | 80.5±0.9 |
| MaPLe | 79.2±1.3 | 82.5±0.8 | 94.6±0.3 | 96.5±0.4 | 95.1±0.4 | 88.8±0.7 | 76.3±0.9 | 48.9±1.5 | 74.6±1.1 | 74.5±1.1 | 81.1±0.9 |
| KgCoOp | 79.9±1.2 | 84.1±0.7 | 94.1±0.4 | 97.5±0.2 | 95.3±0.4 | 84.7±0.8 | 74.1±0.9 | 55.2±1.5 | 72.9±1.1 | 73.9±1.2 | 81.2±0.9 |
| CoCoOp | 79.8±1.2 | 83.4±0.8 | 93.8±0.4 | 97.4±0.3 | 95.4±0.4 | 86.3±0.7 | 76.0±0.9 | 52.2±1.6 | 76.7±1.1 | 74.1±1.2 | 81.5±1.0 |
| AllFT | 80.4±1.3 | 83.5±0.8 | 94.9±0.3 | 96.9±0.3 | 95.4±0.3 | 88.5±0.7 | 76.5±0.9 | 54.8±1.4 | 77.2±1.0 | 78.8±1.0 | 82.7±0.9 |
| VisualFT | 80.0±1.2 | 83.0±0.8 | **95.1**±0.3 | 96.6±0.4 | 95.1±0.4 | **89.9**±0.7 | **78.3**±0.8 | 52.7±1.4 | 80.1±0.9 | 77.7±1.0 | 82.9±0.9 |
| TextFT | **80.9**±1.2 | **85.4**±0.7 | 94.2±0.4 | **98.3**±0.2 | **96.0**±0.3 | 85.6±0.8 | 75.8±0.9 | **62.5**±1.4 | 80.3±0.9 | **79.0**±1.0 | 83.8±0.9 |

models, and (3) compares different transfer algorithms for multimodal models for base evaluation and base-to-novel evaluation.

**Evaluated models and algorithms.** For pretrained models, we evaluate supervised models including ResNet-50 (He et al., 2016), SwinTransformer-base (Liu et al., 2021), ConvNext-base (Liu et al., 2022) trained on ImageNet 1K, and BiT-R101 (Kolesnikov et al., 2020) trained on ImageNet 21K; self-supervised ImageNet models including MAE-base (He et al., 2022), IBOT-ViT-base (Zhou et al., 2022a) and EsViT-Swin-base (Li et al., 2022a); multimodal pretrained model CLIP (Radford et al., 2021) trained on 400 millions image-text pairs; and self-supervised models DINOv2-small, DINOv2-base (Oquab et al., 2023) trained on 142M curated images. For transfer algorithms for pure vision models, we evaluate linear probing (Zhang et al., 2016), Finetune (He et al., 2022), and several parameter-efficient finetuning methods including LoRA (Hu et al., 2022), BitFit (Zaken et al., 2022), SSF (Lian et al., 2022), Adapter (Houlsby et al., 2019), Adaptformer (Chen et al., 2022), VPT (Jia et al., 2022) and TSA (Li et al., 2022b). For transfer algorithms for multimodal models, we evaluate CoOp (Zhou et al., 2022c), CoCoOp (Zhou et al., 2022b), VPT (Jia et al., 2022), MaPLe (Khattak et al., 2023), KgCoOp (Yao et al., 2023), ProGrad (Zhu et al., 2023), Finetune of visual encoder, Finetune of text encoder and Finetune of both encoders. We give results in Table 4-7. We make following observations.

**The size of the pretraining dataset matters.** As seen from Table 4, models trained on ImageNet-1K have very similar performance when well-tuned (except for ResNet-50 which does not use most of the training tricks), regardless of the training algorithm and architecture used. The difference between the worst-performing MAE and best-performing IBOT is 0.6, smaller than the range of

Table 7: Sub-benchmark of FEWTRANS that compares different transfer algorithms for base-to-novel evaluation of multi-modal pretrained models. CLIP-base is chosen as the pretrained model.

| | ImageNet-S | DTD | CIFAR-100 | Flowers | UCF | EuroSAT | Quick Draw | Fungi | Plant Disease | Aircraft | Average |
|---|---|---|---|---|---|---|---|---|---|---|---|
| CoCoOp | 67.5±1.4 | 66.7±1.1 | 86.8±0.5 | 77.5±1.1 | 87.9±0.7 | 67.5±1.3 | 61.4±1.1 | 21.8±1.2 | 59.4±1.4 | 47.9±1.7 | 64.4±1.2 |
| CoOp | 64.3±1.5 | 71.5±1.0 | 86.5±0.6 | 85.0±0.8 | 86.1±0.7 | 71.5±1.2 | **60.9**±1.2 | 31.5±1.4 | 65.0±1.2 | 46.2±1.9 | 66.8±1.2 |
| ProGrad | 65.0±1.5 | 71.7±1.0 | 86.7±0.5 | 85.0±0.8 | 86.3±0.7 | 72.0±1.2 | 61.1±1.2 | 32.5±1.4 | 65.7±1.2 | 49.7±1.8 | 67.6±1.2 |
| VPT | 71.7±1.3 | 67.7±1.0 | **87.5**±0.6 | 84.5±0.8 | 86.4±0.7 | 68.1±1.4 | 56.7±1.2 | 37.0±1.3 | 56.9±1.4 | 61.5±1.3 | 67.8±1.1 |
| MaPLe | 70.4±1.3 | 62.1±1.2 | 88.3±0.5 | 82.4±0.8 | 87.3±0.6 | **77.1**±1.3 | 60.8±1.1 | 34.3±1.3 | 62.2±1.3 | 56.2±1.4 | 68.1±1.1 |
| KgCoOp | 68.9±1.4 | **72.7**±0.9 | 87.0±0.5 | 86.6±0.7 | 87.8±0.7 | 70.6±1.2 | 60.6±1.1 | 33.9±1.4 | 66.7±1.2 | 51.9±1.8 | 68.7±1.2 |
| Zero-shot | 73.9±1.3 | 68.7±1.1 | 86.8±0.5 | 87.0±0.7 | 89.0±0.6 | 69.7±1.4 | 58.1±1.2 | 39.3±1.4 | 59.2±1.2 | 61.5±1.3 | 69.3±1.1 |
| VisualFT | 74.0±1.3 | 69.0±1.0 | 88.3±0.5 | 86.7±0.7 | 89.0±0.6 | 70.2±1.4 | 60.4±1.1 | 38.9±1.4 | 67.5±1.3 | 62.2±1.3 | 70.6±1.1 |
| TextFT | **74.2**±1.3 | 69.8±1.0 | 87.0±0.5 | **87.5**±0.7 | 89.8±0.6 | 72.2±1.4 | 59.9±1.2 | 39.2±1.4 | **70.2**±1.2 | 61.7±1.3 | 71.2±1.1 |
| AllFT | 74.1±1.3 | 69.4±1.0 | 88.1±0.5 | 87.2±0.7 | **89.5**±0.6 | 72.3±1.4 | **60.9**±1.1 | **39.6**±1.4 | 68.9±1.2 | **62.8**±1.3 | **71.3**±1.1 |

confidence interval. However, when the dataset size increases, we see a very clear improvement in few-shot transfer performance.

**CLIP meets problems with uncommon class names.** From Figure 4, we see that CLIP exhibits promising performance on most datasets, but performs badly on Fungi and Plant Disease, two fine-grained datasets whose category names are mostly rare words. This is something like a "text domain shift" which requires significant updates for the text encoder. We expect that such problems can be relieved when the number of shots increases, but for few-shot evluation on these two datasets, only using the visual encoder of CLIP (see Table 5) can be better than using both encoders (see Table 6).

**Visual-only transfer algorithms perform similar.** From Table 5, we can see that except for linear probing, all transfer algorithms for pure visual pretrained models have very similar performance and have intersected confidence intervals. This is in contrast to the benchmark of many-shot transfer learning like VTAB (Zhai et al., 2019), where different transfer algorithms are shown to have significant performance gaps (see Chavan et al. (2023) for example).

**Finetune performs surprisingly well** on all sub-benchmarks for transfer algorithms as shown in Table 5-7, especially for multimodal models. Intuitively, finetuning all parameters of the pretrained model with a few samples should meet overfitting problems. Such a phenomenon needs deeper understanding.

**Are we making progress on few-shot multimodal transfer?** While we observe in Table 6 that all specifically designed transfer algorithms for CLIP perform better than zero-shot baseline in base evaluation, they all perform worse than zero-shot baseline in base-to-novel evaluation in Table 7, different from what some of the methods claimed in their paper with the old benchmarks. In contrast, simple finetune, either finetuning a single encoder or finetuning both, surpasses all these methods in both evaluation settings. This indicates we are *not* making progress in this field and we should rethink what's the thing that leads to real improvement of few-shot multimodal transfer performance.

# 6 CONCLUSION AND FUTURE WORK

We have introduced FEWTRANS, a unified, realistic, rigorous benchmark for evaluating few-shot transferability of pretrained models. Our initial exploration of this benchmark shows that transferring from a better pretrained model trained on a large pretraining dataset seems to be much more important than using a better transfer algorithm. However, we believe that with rigorous evaluation, comparison and further investigations on FEWTRANS, good transfer algorithms will finally emerge. We are now implementing more algorithms and trying to include more pretrained models in the benchmark. In addition to comparing few-shot performance, we plan to add a comparison of the number of tunable parameters and the time needed for a complete adaptation for transfer algorithms.

## 7 REPRODUCIBILITY STATEMMENT

We do our best to ensure the reproducibility of our benchmark. We include most details of our empirical investigations and the benchmark in the two sections of Appendix. The code for the benchmark can be found at `https://anonymous.4open.science/r/FewTrans-7FB5`.

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

## A  EXPERIMENT DETAILS OF SECTION 4

For all empirical investigations in this section, when looking for the optimal hyperparameters for a single task or average optimal hyperparameters for several tasks, we set the overall search space of the number of epochs to be $[1, 2, 5, 10, 15, 20, 30, 40, 50, 60, 70, 80, 90, 100]$. For learning rates, we do not set an upper limit for the search, and for each exponent of 10, we search over its multiples of 1, 2, and 5. For example, for the $1e-05$, we search over $1e-05$, $2e-05$, and $5e-05$. We set the lower limit for the search as $1e-08$. To reduce the search space, we first empirically initialize a coarse-grained search grid. For example, for ViTs, the search space for the number of epochs is $[30, 50, 70]$, while for the backbone learning rate it is $[1e-06, 1e-05, 1e-04]$, and for the head learning rate is $[0.001, 0.01, 0.1]$. We then adjust this coarse-grained search grid such that we can find a coarse-grained "optimal" point inside it. Then we switch to the fine-grained search mode. We start from this coarse-grained point and search over its nearby points to find the "true" optimal one.

For cross-validation on each task, to avoid introducing too much computation overhead, we do not set fine-grained search grid for hyperparameters, and just set a coarse-grained search grid. The determination of the range of the search grid is the same as that of hyperparameter ensemble introduced in Section 5.1.

For hyperparameter configurations in Table 3, we use $1e-06, 1e-05, 1e-04$ as the backbone learning rates for EuroSAT and $1e-07, 1e-06, 1e-05$ for aircraft; we use $0.001, 0.01, 0.1$ as the head learning rates for EuroSAT and $5e-04, 5e-03, 5e-02$ for EuroSAT. The epoch is fixed at 30 for all settings except for lr+epoch ensemble where $13, 26, 40$ are used as epoch candidates.

# B    MORE DETAILS OF FEWTRANS

## B.1    DATASETS

1. **ImageNet-Sketch** (Wang et al., 2019) is a variant of ImageNet (Deng et al., 2009) that contains 50000 sketches of all 1000 classes in ImageNet-1K. The leaves of the wordnet that are reachable from 'carnivore' and 'device' form all 288 classes of novel set and other 712 classes belong to the base set.

2. **DTD** (Cimpoi et al., 2014) is a texture database containing 5640 classes with 47 classes. We randomly choose 37 classes as the base classes, and the other 10 classes as the novel classes.

3. **CIFAR-100** (Krizhevsky et al., 2009) is a real-world dataset that has 100 classes containing 600 images each. We randomly choose 80 classes as the base classes, and the other 20 classes as the novel classes.

4. **VGG Flowers** (Nilsback & Zisserman, 2008) is a dataset of natural images of 102 flower categories. Each category contains 40 - 258 images. We randomly choose 82 classes as the base classes, and the other 20 classes as the novel classes.

5. **UCF-101** (Soomro et al., 2012) is an action recognition data set of realistic action videos, collected from YouTube, having 101 action categories with 13320 videos. We use sampled frames from these videos to form an image dataset. We randomly choose 81 classes as the base classes, and the other 20 classes as the novel classes.

6. **EuroSAT** (Helber et al., 2019) is based on Sentinel-2 satellite images covering 13 spectral bands and consisting of 10 classes with in total 27000 labeled and geo-referenced images. We randomly choose 6 classes as the base classes, and the other 4 classes as the novel classes.

7. **Quick Draw** (Jonas et al., 2016) is a dataset of black-and-white drawings across 345 categories. We use its smaller version used in DomainNet (Peng et al., 2019) that contains 172500 images. We randomly choose 276 classes as the base classes, and the other 69 classes as the novel classes.

8. **Fungi** (Schroeder & Cui, 2018) is a dataset of 1394 categories of mashrooms species, with approximately 100K images. We randomly choose 1115 classes as the base classes, and the other 279 classes as the novel classes.

9. **Plant Disease** (Mohanty et al., 2016) is a dataset that covers 38 categories of plant diseases. We randomly choose 30 classes as the base classes, and the other 8 classes as the novel classes.

10. **Aircraft** (Maji et al., 2013) is a dataset containing 100 aircraft categories, with 100 images each. We randomly choose 80 classes as the base classes, and the other 20 classes as the novel classes.

## B.2    IMPLEMENTATION DETAILS

To determine the range of hyperparameters for hyperparameter ensemble, we first need to find the average optimal hyperparameters on the held-out dataset. This procedure is exactly the same as the procedure we use in Section 4 (see Section A for details). Then for unimodal models that have two learning rates to be tuned, we expand the optimal hyperparameters into a $3 \times 3 \times 3$ grid. For the learning rate, we multiply the optimal hyperparameter found on the held-out dataset by 5 times and then divide it by 5 times to form the axis. For the number of epochs, we add 10 to the optimal hyperparameter, and expand it with its one-third and two-thirds to form the axis. For multimodal models and transfer algorithms that have only one learning rate to be tuned, we expand the optimal hyperparameters into a $5 \times 3$ grid, where the other rules remain unchanged. All points in the obtained grid will be used as hyperparameter configurations for the hyperparameter ensemble.

We conducted all experiments on 16 GeForce GTX 1080 Ti. For each evaluation of 600 tasks, the time cost on one GPU ranges from several hours to several days, depending on the pretrained models and transfer algorithms used. For all algorithms, we do not use weight decay for adaptation, and use default hyperparameters of Adam. For all transfer algorithms, we use their default settings of all other hyperparameters except for learning rates and the number of epochs. We set the batch size of the transfer process to be the maximum number that the GPU memory permits. During the evaluation, we fix the seed all the time, so in fact, the sequence of sampled tasks, and even the sequence of batch sampling inside each task, are exactly the same for all evaluation tasks, which ensures absolutely fair comparison throughout the process.

