# OpenReview forum: "Benchmarking Few-shot Transferability of Pre-trained Models with Improved Evaluation Protocols"
_ICLR.cc/2024/Conference — ICLR 2024 Conference Withdrawn Submission_

### Official Review · Reviewer_byha · 2023-10-18

**Soundness:** 3 good
**Presentation:** 2 fair
**Contribution:** 2 fair
**Rating:** 5
**Confidence:** 4

**Summary:**

The paper proposes a new benchmark, FewTrans, for evaluating few-shot transfer learning of classification tasks. It contains three sub-benchmarks for comparing pretrained models, transfer algorithms for vision models, and transfer algorithms for multimodal models respectively. The authors conducted experiments to evaluate different models and algorithms on the proposed benchmark.

**Strengths:**

- This paper presents a new benchmark for few-shot transfer evaluation and conducts extensive experiments on the proposed benchmark.
- The authors identified some problems in the existing evaluations, e.g. model selection, no class imbalance, choice of hyper-parameters, etc. These insights can be useful for future research. (I have some comments for this point though - see weakness below)
- The paper is overall well-written and easy to follow.

**Weaknesses:**

- There are a few more existing benchmarks for few-shot evaluation besides Meta-Dataset the authors mentioned. The authors should compare the proposed benchmark with them in terms of the basic data statistics, coverage of domains, sizes, etc., to give a clearer impression of the difference. E.g., Meta-Album (Ullah et al, 2022), Meta-Omnium (Bohdal et al, 2023). These datasets also contain various sub-datasets. The authors should better explain what is the unique advantage of the proposed FewTrans.
- One of the claims the authors made is that the datasets previous studies used are too easy and lack diversity. However, from table 5, the performance of different transfer learning methods is very close, which can somehow indicate the proposed benchmark also lacks diversity and the performance on some datasets can be > 90%, which also means they are not difficult enough.
- It would be good to have various numbers of shots in the benchmark - currently there's only 10-shot. I think the performance of different transfer methods may be different between, e.g., 5-shot and 10-shot.
- Why didn't the authors also evaluate some meta-learning algorithms since it's a very commonly used strategy for few-shot transfer learning and has proven to be very effective, at least for vision-only models?

Minor:
The authors should state somewhere early in the paper that this benchmark actually focuses on classification instead of just saying "few-shot transfer".

**Questions:**

See above.

---

### Official Review · Reviewer_rvMz · 2023-10-29

**Soundness:** 2 fair
**Presentation:** 3 good
**Contribution:** 3 good
**Rating:** 5
**Confidence:** 5

**Summary:**

The paper present a benchmarking protocol to study few-shot recognition. The paper first analyzes limitations of existing protocols to motivate the new one, e.g., from results in different confidence intervals, the size of validation sets for hyperparameter tuning, large variation in few-shot regimes, etc. It also embraces pretrained models. With the new protocol, the paper compares existing few-shot learning methods and leads to several conclusions, e.g., we are not making progress in this field".

**Strengths:**

The paper has several merits.

- The motivation is good. It does a good job in terms of analyzing the limitations of current protocols in the few-shot recognition field.
- The presented new protocol makes sense in general that addresses the aforementioned limitations.
- The experiments are sufficient that it studies representative few-shot recognition methods.

**Weaknesses:**

Below are several concerns related to weaknesses.

- In Introduction, the paper explains few-shot learning is "facilitating downstream scenarios where labeled data can be expensive or difficult to obtain". While few-shot learning is somewhat established in the community, can authors motivate the study of few-shot learning with real-world applications? When factually does an application need few-shot learning?

- The paper writes "sampling class-imbalanced tasks". As the few-shot learning setting has only a few examples for each class, how to set a reasonable class-imbalanced task? Can authors explain with concrete details?

- The paper writes "Thus to make the evaluation protocol realistic while being fair for comparison, we have two choices: (1) determine hyperparameters of transfer algorithms in advance on a held-out dataset that is both different from the pretraining dataset and target downstream dataset". It is not clear why must it be different from both pretraining and target datasets. When embracing a pretrained model (e.g., CLIP's visual encoder), how to define such a validation set different from the pretraining data? Hypothetically, it is reasonably to think the pretraining data reflects data in the real world already. Can authors clarify and explain?

- While the paper makes a good point "a good few-shot transfer method should not only have high performance at its optimal hyperparameters, but should also have resistance to the change of hyperparameters," why is being resistant to different hyperparameters a reasonable argument in designing a good few-shot setup? Can we leave hyperparameter search as a standalone method/problem orthogonal to a few-shot protocol?

- The paper writes "No variation of the number of classes." But it is not clear why "the number of classes" matters. Can authors explain and clarify?

- In Section 5.1, the paper suggests "hyperparameter ensemble". But why don't we leave hyperparameter search as a algorithmic problem in few-shot learners? What if a few-shot method has good hyperparameters outside the predefined range of hyperparameters?

- The paper writes "every evaluated task can provably approach its optimal performance" but doesn't provably justify this claim. Authors should clarify.

- The paper writes "the hyperparameters won’t usually change too much from dataset to dataset". This is not clear. Can authors justify this?

- The paper writs "We thus restrict the maximum number of training samples in each class to 10, constructing “true” few-shot tasks." However, this conflicts previous suggestions of setting "class-imbalanced tasks". Can authors explain? Moreover, why 10 examples per class construct "true few-shot tasks"? Can authors discuss what numbers of per-class examples can be grounded as "true few-shot tasks"?

- While the paper tries to embrace pretrained models such as CLIP, it is not clear what few-shot tasks can be reasonable given that pretrained models have seen examples of few-shot tasks (e.g., images belonging to ImageNet classes). Can authors clarify?

- The paper writs "Fungi and Plant Disease, two fine-grained datasets whose category names are mostly rare words." How to tell the categories in these datasets are "mostly rare words"? Can authors clarify?

- The paper writes "This is something like a “text domain shift"". Can authors explain what it means? Shift from where to where?

- The paper has a strong claim that "This indicates we are not making progress in this field and we should rethink what’s the thing that leads to real improvement of few-shot multimodal transfer performance." This is not an insightful claim and I'd encourage authors to write more insightful comments, e.g., how to move forward with the proposed benchmark, what directions are reasonable to move forward, etc.

**Questions:**

Questions are in the weaknesses. I encourage the authors to address them in rebuttal.

---

### Official Review · Reviewer_m8iP · 2023-11-04

**Soundness:** 1 poor
**Presentation:** 2 fair
**Contribution:** 3 good
**Rating:** 1
**Confidence:** 4

**Summary:**

The paper proposes a new benchmark for few-shot transfer learning. This addresses deficiencies in the prior benchmarks, for example, small sample size in the task sets leading to unreasonably inflated confidence intervals and non-unified model selection process based on hyperparameter search.

**Strengths:**

- The need in a robust and fair benchmark for few-shot learning is imminent and the paper is addressing an important topic
- Section 4.2 is spot on. The concept of one single held-out dataset for tuning model hyperparameters seems a very good way of organizing the benchmark and making sure that the model is tested across a variety of inference datasets without hyperparameter modifications. This is also a very good way of minimizing the leakage of data from the test dataset to the model. However, the structure of the dataset is scattered across Section 5, making the description a bit vague. For example, the fact that  for ImageNet models, CUB is used as held-out dataset is only mentioned in passing around Table 4. I believe this is one of the key benchmark components. I suggest that the authors provide a clear description of the benchmark workflow somewhere in the text in one spot (maybe visual depiction could help here?). I basically want to see all the steps: backbone pretraining (which datasets and which tasks?) -> hyperparameter fine-tuning (which datsets, tasks?) -> few-shot/zero-shot inference (which datasets and tasks?). So far, it looks like one may need to read the paper a few times to get all the pieces from different locations in one's mind. Is there a more effective way of communicating this information in one shot (no pun intended ;))

**Weaknesses:**

- The contributions of the paper are not explicitly provided as a bullet-point list. Please include in the revised version
- In Section 4.1, the paper discusses only problematic cases in the existing few-shot benchmarks literature, especially in the case of task sample size. It may create an impression that there have not been any reliable few-shot results in the literature due to all existing benchmarks being compromised. I doubt that this is true and I also doubt that this is the message that the authors want to send by writing this paper. I believe that in the regard of task sample size and metric reliability, there are plenty of reasonable benchmarks available in prior work such as miniImagenet (from https://arxiv.org/abs/1606.04080), tieredMiniImagenet (from https://arxiv.org/abs/1803.00676), FC-100 (from https://arxiv.org/abs/1805.10123) and of course Meta-Dataset (from https://arxiv.org/pdf/1903.03096.pdf). It is necessary to extend the prior work discussion accordingly, either in related work section or Section 4.1, to make the exposition more balanced.
- In terms of zero-shot datasets, CUB (https://authors.library.caltech.edu/records/cvm3y-5hh21) and FLOWERS (https://ieeexplore.ieee.org/document/4756141) have been driving the algorithm development for significant amount of time. On a related note, a few problems have been identified in the literature related to setting up robust multi-modal benchmarks https://arxiv.org/pdf/1707.00600.pdf. It would be good to have this reflected in the literature review. I have one specific question related to this line of work. One significant issue with few- and zero-shot operation happens when both unseen and seen classes are mixed at inference time, which is more than realistic scenario, see e.g. (https://arxiv.org/pdf/1906.11892.pdf). How is this addressed in your benchmark?
- It may well be that it might be acceptable to chose one of the datasets as held-out dataset for hyperparameter tuning. However, the assumption here is that results do not change depending on the chosen held-out dataset. I believe that authors may fall here in the same trap of noise-driven model ranking as they so keenly identified in some other benchmarks. I understand that it might be overkill to require people using the benchmark to do leave-k-out cross-validation on held-out hyperparameter datasets. But it actually provides the authors with the opportunity to make a more significant scientific contribution in the methodology of benchmark design. In my mind, if authors could prove that it is valid to use only CUB as held-out dataset, it could actually be a significant contribution. Basically, for the sake of benchmark design, can you present evidence that the ranking of models produced by your benchmark does not change if you use FLOWERS and a few other datasets as held out datasets, comparing this to the CUB-based result? With absence of evidence like this, I see significant risks of this benchmark falling in exactly the same methodological trap as the other benchmarks criticized in the paper. So far, the evidence presented in paragraph "Optimal hyperparameters change from dataset to dataset" of Sections 4.2 basically provides the evidence to support this concern.
- Paragraph "Cross-validation fails to provide reliable estimation of hyperparameters" in Section 4.2 seems to be disconnected from the idea of constructing a robust few-shot benchmark. It certainly makes sense that cross-validation on 5-shots of data is not a very fruitful direction in optimizing learning algorithm generalization. However, if this is not a good strategy, it should show up in accuracy results reported by a good benchmark, right? Otherwise learning algorithm designers should be free to use any methodologically sound (for example, no test data leakage into trining procedure) techniques to achieve best score on a benchmark. How is this connected to the benchmark itself - it not at all clear? I would suggest removing this paragraph altogether from the methodology part of the paper. If authors feel that this is a valuable discovery, maybe it may belong to empirical results section or could be migrated to appendices?
- Section 4.3, paragraph "No variation of the number of classes". I don't understand what authors wanted to say here. Please rewrite the paragraph to make the message clear.
- Section 5.1. Ensembling is a well known and time-tested ML technique. However, this technique also requires one or two orders of magnitude more compute at inference time than the usual inference based on a single model. I do not see how this is making the proposed benchmark more fair and robust. When we use an ensemble, it is actually better if every single model in ensemble is overfitting more and it is more sensitive to hyperparameter perturbations. Whereas Section 4.2 seems to argue that a benchmark should favour models that are more stable. It feels like this part of the paper is saying that every model should run in production as an ensemble. This seems like the benchmark in this part is emposing a certain algorithmic solution for learning algorithms. More importantly, a lot of the time the only possibility is to run a single model at inference due to computational limitations (such as real-time for example). So models whose individual instances are actually more stable to hyperparameter variations and therefore have less potential to get additional boost from ensembling will be rejected by the proposed benchmark in favour of less stable models that generalize well only when combined in ensemble. I believe this is a serious methodological gap in this proposed benchmark.
- How do you make sure that the backbone pretraining datasets and the test datasets are truly disjoint? For example, https://arxiv.org/pdf/1707.00600.pdf made a great deal of effort in making sure that the images and classes that are used in training the ImageNet backbone are disjoint from the test cases, going as far as releasing backbone weights pretrained on a specific subset of ImageNet. tieredIMageNet and FC-100 make class splits according to higher-level class hierarchies to make sure that similar classes do not end up in both training and test data. How do you deal with this serious issue? I do not find enough information about it in the paper. Paragraph "Evaluated models and algorithms." on page 8 seems to suggest that you do not pay special attention to enforcing a rigorous split. This poses a grave data leakage risk.
- "Visual-only transfer algorithms perform similar.", "Finetune performs surprisingly well" "Intuitively, finetuning all parameters of the pretrained model with a few samples should meet overfitting problems. Such a phenomenon needs deeper understanding." I argue that all of the above is the artifact of the proposed benchmark, and specifically the ensembling part of it. Basically, finetuning indeed produces overfitting models, but ensembling makes a more powerful model from these overfitting models at the expense of more computationally demanding inference.
- "Are we making progress on few-shot multimodal transfer? ... In contrast, simple finetune, either finetuning a single encoder or finetuning both, surpasses all these methods in both evaluation settings. This indicates we are not making progress in this field and we should rethink what’s the thing that leads to real improvement of few-shot multimodal transfer performance" Subject to all my previous comments, there is enough evidence to suggest that this can be a bold and misleading claim. I suggest that the authors make all my comments into account, rectify methodological gaps and see if they still want to make this claim based on revised experimental results.

**Questions:**

- Page 3: typo: adptation -> adaptation
- Related work and few-shot transfer learning problem definition can be made subsections in the Intro to save space - sections are fairly small.
- "Because of this difficulty, a good few-shot transfer method should not only have high performance at its optimal hyperparameters,
but should also have resistance to the change of hyperparameters, that is, the test loss landscape around the optimal hyperparameters should be flat such as we can tolerate an inevitable deviation of hyperparameter estimations". 100% agree with the statement. The main question is how does your benchmark makes sure that it favours algorithms with this property? Can you explain the intuition or theoretical mechanism you implement in your benchmark to achieve this goal? Can you present evidence that your benchmark favours algorithms with this property?
- Page 5: typo "papers that evaluates" -> "papers that evaluate"
- "which has proven to have tons of mislabeled images and outliers". "Tons" is not a scientifically appropriate quantifier of the number of mislabelled images. Could you please be more specific and data driven? Isn't it viable to cite the percentage or the absolute number of mislabelled images from the original study?
- "we sample 600 tasks per dataset and report the 95% confidence intervals." Can you describe the exact procedure used to compute confidence intervals?
- What is the significance of the study in Table 4 discussed in paragraph "The size of the pretraining dataset matters.". With the lack of control over training and test set overlap that I flagged above, the generalization improvement can have 2 root causes. First, data size increase produces a better generalizing model. Second, the data size increase implies larger overlap between train and test. In the second case, the accuracy improvement may be a result of simply having a less pure few-shot scenario caused by significant overlap between training and test data content. Can you present the evidence that the likelihood of Scenario 1 prevails over that of Scenario 2?
- "CLIP meets problems with uncommon class names. From Figure 4, we see that CLIP exhibits
promising performance on most datasets, but performs badly on Fungi and Plant Disease" There is no Figure 4 in the paper.
- The setup used to derive results in Tables 5 and 6 needs to described in detail. Not clear at all how CLIP is used and what the relationship is between CLIP and the rows in the Tables.

- I outlined a few questions and concerns that require attention and if all of them are addressed meticulously I will be more than happy to revise my score accordingly.

---

### Official Review · Reviewer_QQrt · 2023-11-06

**Soundness:** 3 good
**Presentation:** 3 good
**Contribution:** 2 fair
**Rating:** 5
**Confidence:** 4

**Summary:**

The authors present a benchmark for fewshot tranfer learning. The motivation behind introducing a new benchmark is mainly that a good benchmark does not exit. The authors highlight two mistakes in evaluation of fewshot transfer elarning methods in literature,
1. model selection is difficult and subject to bias
2. high variance in performance because the performance can be largely influenced by the choice of sampled data. This is more significant in fewshot settings.

**Strengths:**

Evaluation of fewshot learning algorithms is a challenge. The fact that with so many papers on the topic , there is not a consesus benchmark attests to this problem.
The paper identifies an important problem. Robustness with respect to hyperparameters are mostly overlooked in transfer learning. One way to take that into account is to  marginalize performance over hyperparameters, which this paper aims to do.

The paper is generally easy to follow and well written.

**Weaknesses:**

the authors talk about having diverse dataset but yet similar data from most of these dataset can be found in imagenet.
real world application of fewshot transfer learning is on out of distribution generalization and it would be nice to have an axes in this bench mark to see to what degree of dataset shift a given model can generalize to. to that end it would be good to have some in domain dataset, some near domain and some far domain datasets. see paper on paper on extreme dataset shift and that from fereshteh.

lack of medical domain datasets even thought transfer learning is the main used tool in medical image analysis.

the paper mentions that " Since we know that the hyperparameters won’t usually change too much from dataset to dataset, we determine the range on a held-out dataset by finding the best average optimal hyperparameters on this dataset, setting it as the center of the hyperparameter range, and expanding it to a full range". There are various papers suggesting that deep learning methods are very under specified when distribution changes. This under-specification extends to heyperparameters too. meaning that the hyperparameters would be different, especially when the data examples are few. I am not convinced that having a held out dataset to fine tune hyper-parameters on would be robust and would generalize to other datasets. While this method may be less bad when the helout dataset is close to the test datasets (like what is done in unsupervised domain adaptation ), I do not think it should be considered as a universal approach to evaluate generalization. This approach would require extensive experimentation to provide emperical evidence.

**Questions:**

I would ask the reviewers about the justification of the statement i mentioned above and also brought in the following, that where they have done any experiments to prove it or if they can prove it theoretically.
Since we know that the hyperparameters won’t usually change too much from dataset to dataset, we determine the range on a held-out dataset by finding the best average optimal hyperparameters on this dataset, setting it as the center of the hyperparameter range, and expanding it to a full range